# Non-Laboratory-Based Risk Prediction Tools for Undiagnosed Pre-Diabetes: A Systematic Review

**DOI:** 10.3390/diagnostics13071294

**Published:** 2023-03-29

**Authors:** Will Ho-Gi Cheng, Yuqi Mi, Weinan Dong, Emily Tsui-Yee Tse, Carlos King-Ho Wong, Laura Elizabeth Bedford, Cindy Lo-Kuen Lam

**Affiliations:** 1Department of Family Medicine and Primary Care, Li Ka Shing Faculty of Medicine, The University of Hong Kong, Hong Kong, China; 2Department of Family Medicine, The University of Hong Kong Shenzhen Hospital, Shenzhen 518009, China; 3Department of Pharmacology and Pharmacy, Li Ka Shing Faculty of Medicine, The University of Hong Kong, Hong Kong, China

**Keywords:** pre-diabetes, early detection, risk prediction tools, non-laboratory-based

## Abstract

Early detection of pre-diabetes (pre-DM) can prevent DM and related complications. This review examined studies on non-laboratory-based pre-DM risk prediction tools to identify important predictors and evaluate their performance. PubMed, Embase, MEDLINE, CINAHL were searched in February 2023. Studies that developed tools with: (1) pre-DM as a prediction outcome, (2) fasting/post-prandial blood glucose/HbA1c as outcome measures, and (3) non-laboratory predictors only were included. The studies’ quality was assessed using the CASP Clinical Prediction Rule Checklist. Data on pre-DM definitions, predictors, validation methods, performances of the tools were extracted for narrative synthesis. A total of 6398 titles were identified and screened. Twenty-four studies were included with satisfactory quality. Eight studies (33.3%) developed pre-DM risk tools and sixteen studies (66.7%) focused on pre-DM and DM risks. Age, family history of DM, diagnosed hypertension and obesity measured by BMI and/or WC were the most common non-laboratory predictors. Existing tools showed satisfactory internal discrimination (AUROC: 0.68–0.82), sensitivity (0.60–0.89), and specificity (0.50–0.74). Only twelve studies (50.0%) had validated their tools externally, with a variance in the external discrimination (AUROC: 0.31–0.79) and sensitivity (0.31–0.92). Most non-laboratory-based risk tools for pre-DM detection showed satisfactory performance in their study populations. The generalisability of these tools was unclear since most lacked external validation.

## 1. Introduction

In 2021, type 2 diabetes mellitus (T2DM) accounted for up to 6.7 million deaths, while impacting the lives of 537 million individuals globally [1]. T2DM is often preceded by a stage of sub-DM hyperglycaemia, known as pre-diabetes (pre-DM), which lasts for several years and can be reversible [2]. Indeed, with timely intervention, the blood glucose levels of pre-DM individuals can return to within the normal range [3]. Therefore, cost-effective methods that use the clinical and/or anthropometric characteristics of individuals to predict their pre-DM risks have gained a lot of attention among researchers and clinicians. Such methods can include risk prediction tools, models or algorithms. 

To our knowledge, only one review, published in 2014, has evaluated pre-DM risk tools and included studies up to 2013 [4]. The review found that existing pre-DM tools offered similar internal predictive performances despite varying development methods and different numbers of predictors included [4]. However, it is important to note that the majority of studies included in this review used laboratory biomarkers (e.g., blood triglyceride levels) as predictors in the models [4], which limits the applicability for case-finding in general and primary care populations. Pre-DM risk prediction tools are intended to be simple, low-cost and non-laboratory-based in order to save unnecessary blood tests. The inclusion of laboratory biomarkers cannot be cost-effective as the amount of time and cost incurred for an individual to obtain the required laboratory variable would be similar to performing a pre-DM and DM diagnostic blood test directly. Notably, recent studies have reported the association between DM risks and other less common, modifiable lifestyle factors, e.g., the level of alcohol consumption [5] and the quantity of sleep [6]. As a result, there has been increasing attention on using modifiable predictors to develop risk prediction tools. For instance, despite being developed by different methods, both of the non-laboratory-based risk prediction tools developed by Dong et al. in 2022 included sleeping hours as one of the predictors [7], which could indicate the clinical and statistical significance of such predictors in predicting pre-DM risks. Having said that, the effects of such lifestyle predictors on the prediction accuracy and performance of non-laboratory-based pre-DM risk prediction tools has not been reviewed. Furthermore, due to recent technological advancements, a number of recent studies that used novel methods, such as artificial intelligence and machine learning (ML), to develop prediction tools have been published since Barber et al.’s 2014 review [4]. 

The current study therefore aimed to systematically review existing non-laboratory-based pre-DM tools published in the literature, focusing on identifying important non-laboratory predictors and evaluating the performance of these tools to provide an update on the current evidence.

## 2. Materials and Methods

### 2.1. Search Strategy

Separate searches were conducted on three medical databases (PubMed, Embase, MEDLINE), and on one nursing-related database (CINAHL), from 1946 until February 2023 to identify available studies. Embase and MEDLINE were searched via Ovid, while CINAHL was searched via EBSCOhost. In order to avoid missing potential studies, citation searching on reference lists of selected studies, and internet manual searching on Google Scholar were conducted. The detailed search strategy is listed in Appendix A. 

### 2.2. Screening and Selection of Studies

Studies were included if they met all of the following criteria: Included pre-DM as the only, or one of the, main outcome(s) of the risk prediction tool;Reported the main outcome using: (i) fasting glucose, (ii) 2-h post-prandial glucose, or (iii) haemoglobin A1c (HbA1c);Provided a detailed methodology for the development of their tool;Only utilised non-laboratory predictors as their prediction variables;Developed tools that were for adults (≥18 years old) in the general population;Published in the English language with full-text available.

Conversely, studies were excluded if they met any of the following:
Included gestational DM or Type 1 DM as the outcome(s) of risk prediction;Only investigated associations between predictors and outcomes;Only aimed to develop or test theoretical algorithms without the intention of implementation in clinical practice;Utilised any laboratory or genetic predictors as their prediction variables;Developed the tool for a specific population, e.g., pregnant women, children, patients of a specific disease group, or older people;Commentaries, editorials, conference abstracts, and systematic reviews.

EndNote X9 and EndNote 20 were used to store and manage identified studies. Following the removal of duplicates, two reviewers (W.C. and Y.M.) independently screened the titles and abstracts to select eligible studies based on the inclusion and exclusion criteria. Full texts of selected studies were then retrieved and independently reviewed. Disagreements or discrepancies were resolved through discussion to reach an agreement between the two reviewers.

### 2.3. Data Extraction and Quality Assessment

Data from the selected studies were extracted and tabulated into a Google Spreadsheet for the narrative synthesis, according to the following list: (1) study region, (2) study sample size, (3) data source for the study sample, (4) prediction outcome and its measurements, (5) methods used for tool development, (6) methods used for predictors selection, (7) predictors included in the final tool, and (8) performance evaluation measures in internal and/or external validation, including area under the Receiver-Operating Characteristic curve (AUROC), sensitivity, specificity, positive predictive value (PPV), and negative predictive value (NPV), with their respective 95% confidence intervals if reported. For ease of interpretation, extracted predictors were categorised into three groups: (i) socio-demographic factors, (ii) clinical factors, and (iii) lifestyle factors. Furthermore, predictors of a similar nature were combined under one broad umbrella term. For instance, (i) the predictor variable “hypertension” in this review included use of antihypertensive medications, history of hypertension, and duration of hypertension, but excluding ‘systolic/diastolic blood pressure levels’ as ‘blood pressure’ was counted as a standalone predictor; (ii) “history of hyperglycaemia” referred to past episodes of hyperglycaemia confirmed by a blood test in a medical check-up, during an illness, or during pregnancy; (iii) “dyslipidaemia” included dyslipidaemia and history of hyperlipidaemia; and (iv) “family history of DM” summarised any predictors related to the number of parents and/or siblings with DM. All units were converted to mmol/L for blood glucose levels, and to percentage for HbA1c for comparison and consistency. 

We applied the Clinical Prediction Rule Checklist of Critical Appraisal Skills Programme (CASP) appraisal checklist [8] to assess the quality and risk-of-bias of the selected studies. This review was reported in compliance with the PRISMA Checklist and PRISMA flowchart [9]. 

Details of the protocol for this systematic review were registered on PROSPERO and can be accessed at www.crd.york.ac.uk/prospero/display_record.php?RecordID=345706 (accessed on 23 August 2022).

## 3. Results

A total of 6398 titles were identified from the database searches. Following removal of duplicates, 4686 articles were screened based on titles and abstracts, and 77 full texts were then retrieved. A total of 19 studies were eligible to be included in the review. Seven additional studies were identified through citation and internet manual searches, with five of them meeting the inclusion criteria. Finally, 24 studies were included in our review (Figure 1). From the 24 studies, there were a total of 28 risk prediction tools developed. Table 1 provides a summary on the study subject characteristics and prediction tools of the included studies. 

The majority of the studies were conducted in Asia (37.5%) [7,10,11,12,13,14,15,16,17], six in the Middle East and North Africa (25.0%) [18,19,20,21,22,23], five in Europe [24,25,26,27,28], three in North America [29,30,31], and one in South America [32]. For the data source of the study sample, about half of the studies (54.2%) developed their risk prediction tools using retrospective population-level health data [7,11,12,15,16,17,18,22,25,26,28,29,31], while the remaining studies sampled their data from community-based health surveys [10,13,14,19,20,21,23,24,27,30,32]. The median development sample size was 2073 and ranged from 308 to 40,381.

### 3.1. Quality of Included Studies

The CASP Clinical Prediction Rule Checklist was applied to assess the quality of the included studies (Appendix A). The validity of the results reported in several studies is uncertain due to the lack of external validation [7,13,16,18,19,20,21,23,28,30,31,32]. As a result, the applicability of the findings is compromised. Furthermore, one study used a small external sample of 83 individuals to validate their tool [14], which could lead to potentially biased results. Overall, it was found that the methods used to construct different tools were adequately reported in nearly all of the studies. However, in a study that developed the prediction tool by ML [22], there could be selection bias due to limited explanations regarding how the factors were selected and weighted in the prediction algorithms. The majority of studies reported the performance of their prediction tools by AUROC, sensitivity, specificity, PPV, and NPV. However, two did not report the AUROC [27,28], four did not report the PPV and NPV of the tools [12,14,17,20], and two reported AUROC without referring to any other performance measurements [10,29]. Ten studies (41.7%) sought to improve the precision of the predictive performances by refining their tools with the addition and/or elimination of predictors following the initial validation [7,12,14,20,25,26,27,28,29,32]. 

### 3.2. Outcomes of Risk Prediction Tools

Eight studies (33.3%) developed tools for predicting pre-DM risk only [11,12,13,14,15,17,18,28] with the remaining sixteen studies focusing on the prediction of both pre-DM and DM risk [7,10,16,19,20,21,22,23,24,25,26,27,29,30,31,32]. An inconsistency in the outcome definition and measure of pre-DM was noted among the studies. For instance, thirteen studies (54.2%) defined pre-DM as when one of the two biochemical parameters were met (fasting/postprandial plasma glucose level (*n* = 11), fasting plasma glucose level/HbA1c (*n* = 1), random plasma glucose level/HbA1c (*n* = 1)) [7,11,12,13,14,16,17,21,22,26,27,30,32]. One third of the studies (37.5%) used only one parameter (fasting plasma glucose level (*n* = 5), postprandial plasma glucose.

**Table 1 diagnostics-13-01294-t001:** Summary of study subject characteristics and prediction tools of the included studies (*n* = 24).

	Development Sample				Outcome of the Tool			Predictors of the Tool				Article Quality
Author, Year	Country/ Region	*n*	Age (Range/ Mean)	Data Source	Extent of Hyperglycaemia	Outcome Measured by	Definition(s)	Development Method(s)	No. of	Predictors Included		No. of CASP Criteria Met (Out of 11)
Abbas, 2021 [18]	Qatar	5814	40.6	Population-based BioBank data	PDM	HbA1c	5.7-6.4%	Multivariate LR model	5	- Age - BMI - HTN - Sex - WC		8
Bahijri, 2020 [19]	Saudi Arabia	1403	32.0	Cluster sampling in healthcare centres	PDM/DM	HbA1c/ FPG/ 1-h PG	≥5.7% ≥6.1 mmol/L ≥8.6 mmol/L	Multivariate LR model	5	- Age - Sex - WC - Hx of HG - Family Hx of DM		8
Barengo, 2017 [32]	Colombia	2060	47.2	Age-stratified sampling among population-wide insurance users	PDM/DM	FPG/ 2-h PG	≥5.6 mmol/L ≥7.8 mmol/L	Multivariate LR model	4	- Age - WC - HTN - Family Hx of DM		9
Dong, 2022 [7]	Hong Kong	1238	40.7	Population-based health survey data	PDM/DM	HbA1c/ FPG	≥5.7% ≥6.1 mmol/L	Multivariate LR model; Extreme Gradient Boosting ML model	7; 8	LR: - Age - BMI - WHR - Smoking - Sleep hours - Exercise - Fruit consumption.	ML: - Age - BMI - WHR - SBP - WC - Smoking - Sleep hours - Exercise	10
Fu, 2014 [10]	China	7953	56.4	Community-based health screening study	PDM/DM	2-h PG	≥7.8 mmol/L	Multivariate LR model	9	- Age - Height - BMI - WC - SBP - Pulse - HTN - DLP - Family Hx of DM		9
Fujiati, 2017 [11]	Indonesia	21,720	>18	Population-based health survey data	PDM	FPG/ 2-h PG	5.6–6.9 mmol/L 7.8–11.0 mmol/L	Multivariate LR model	8	- Age - Sex - Education level - Family Hx of DM - Smoking - Exercise - BMI - HTN		9
Gao, 2010 [12]	China	1986	52.7	Population-based health survey data	PDM	FPG/ 2-h PG	6.1–6.9 mmol/L 7.8–11.0 mmol/L	Multivariate LR model	3	- Age - WC - Family Hx of DM		10
Gray, 2010 [26]	UK	6186	57.3	Population-based screening study data	PDM/DM	FPG/ 2-h PG	≥6.1 mmol/L ≥7.8 mmol/L	Multivariate LR model	7	- Age - Ethnicity - WC - BMI - Sex - Family Hx of DM - HTN		10
Gray, 2012 [25]	UK	6390	57.3	Population-based screening study data	PDM/DM	FPG/ 2-h PG/HbA1c	≥6.1 mmol/L≥7.8 mmol/L≥6.5% ^†^	Multivariate LR model	6	- Age - Ethnicity - BMI - Sex - Family Hx of DM - HTN		10
Gray, 2013 [24]	Portugal	3374	51.5	Cluster sampling in healthcare centres	PDM/DM	FPG	≥5.6 mmol/L	Multivariate LR model	4	- Age - BMI - Sex - HTN		9
Handlos, 2013 [20]	Middle East and North Africa	6588	44.3	Opportunity sampling in study locations	PDM/DM	HbA1c	≥6.0%	Multivariate LR model	7	- Age - BMI - Sex - Family Hx of DM - Family Hx of DM (2) ^‡^ - Hx of GDM - Ethnicity		8
Henjum, 2022 [23]	Algeria	308	≥18	Opportunity sampling in study locations	PDM/DM	HbA1c	≥5.7%	Multivariate LR model	3	- Age - BMI - WC		8
Hische, 2010 [27]	Germany	1737	52.1	Opportunity sampling in healthcare centres	PDM/DM	FPG/ 2-h PG	≥6.1 mmol/L ≥7.8 mmol/L	Decision tree guided by ML	2	- Age - SBP		9
Koopman, 2008 [29]	USA	4045	20–64	Population-based health survey data	PDM/DM	FPG	≥5.6 mmol/L	Multivariate LR model	6	- Age - BMI - Sex - Family Hx of DM - Pulse - HTN		10
Memish, 2015 [21]	Saudi Arabia	1435	≥20	Geographically stratified sampling in healthcare centres	PDM/DM	FPG/ 2-h PG	≥5.6 mmol/L ≥7.8 mmol/L	Multivariate LR model	4	- Age - Hx of GDM - HTN - WC		7
Rajput, 2019 [13]	India	892	42.2	Opportunity sampling in study locations	PDM	FPG/ 2-h PG	5.6–6.9 mmol/L 7.8–11.0 mmol/L	Multivariate LR model	4	- Age - Family Hx of DM - Waist-to-height ratio - DBP		8
Robinson, 2011 [30]	Canada	4366	40–70	Opportunity sampling in community clinics	PDM/DM	FPG/ 2-h PG	≥6.1 mmol/L ≥7.8 mmol/L	Multivariate LR model	12	- Age - BMI - WC - Exercise - Fruit/Veg consumption. - HTN - Hx of HG - Family Hx of DM - Sex - Ethnicity - Macrosomia - Education level		8
Sadek, 2022 [22]	Qatar	1660	37 (median)	Population-based BioBank data	PDM/DM	HbA1c/ RPG	≥5.7% ≥7.8 mmol/L	Multivariate LR model; 4 ML models using: (1) Random Forest, (2) Gradient Boosting Machine, (3) XgBoost, (4) Deep Learning	7	- Age - Sex - WHR - BMI - HTN - DLP - Education level		8
Stiglic, 2018 [28]	Slovenia	2073	54.9	Population-wide electronic medical record dataset	PDM	FPG	6.1–6.9 mmol/L	Multivariate LR model	6	- Age - Sex - WC - Hx of HG - Family Hx of DM - HTN		9
Tan, 2016 [14]	Japan	1054	Not reported	Community-based health screening study	PDM	FPG/ 2-h PG	6.1–6.9 mmol/L 7.8–11.0 mmol/L	Multivariate LR model	5	- Sex - WC - HTN - Hx of HG - Exercise		9
Wang, 2015 [15]	South China	6197	51.6	Population-based health survey	PDM	FPG	6.1–6.9 mmol/L	Multivariate LR model	5; 4	Men: - Age - WC - BMI - Family Hx of DM - HTN	Women: - Age - WC - BMI - Family Hx of DM	10
Xin, 2010 [16]	Rural China	LR: 1131 Tree: 893	52.4	Population-based health survey	PDM/DM	FPG/ 2-h PG	≥6.1 mmol/L ≥7.8 mmol/L	Multivariate LR model; Classification tree analysis	6; 5	LR: - Age - BMI - WHR - Family Hx of DM - HTN - HTN (2) ^§^	Tree: - WHR - WC - HTN - Age - Family Hx of DM	8
Yu, 2010 [31]	USA	3932	≥20	Population-based health survey	PDM/DM	FPG	≥5.6 mmol/L	Multivariate LR model; Support vector machine by ML	10	- Age - Sex - Family Hx of DM - Ethnicity - Weight - Height - WC - BMI - HTN - Exercise		8
Yu, 2022 [17]	China	40,381	44.0	Population-based health survey	PDM	FPG/ 2-h PG	6.1–6.9 mmol/L 7.8–11.0 mmol/L	Multivariate LR model	6	- Age - Education level - Family Hx of DM - WC - BMI - SBP		9

Note: BMI = Body Mass Index, BP = Blood Pressure, consumpt. = consumption, DBP = Diastolic Blood Pressure, DLP = Dyslipidaemia, DM = Diabetes Mellitus, FPG = Fasting Plasma Glucose, GDM = Gestational Diabetes Mellitus, HG = Hyperglycaemia, HTN = Hypertension, Hx = History, LR = Logistic Regression, ML = Machine Learning, PDM = Prediabetes Mellitus, RPG = Random Plasma Glucose, SBP = Systolic Blood Pressure, Veg = Vegetable, WC = Waist Circumference, WHR = Waist-Hip-Ratio, 1-h PG = 1-h Post-prandial Plasma Glucose, 2-h PG = 2-h Post-prandial Plasma Glucose. ^†^ PDM was defined by FPG and 2-h PG while DM was defined by FPG, 2-h PG and HbA1c. ^‡^ The tool included two separate predictors related to family history of diabetes, which were parental history of diabetes and number of siblings with diabetes. ^§^ The tool included two separate predictors related to hypertension, which were history of hypertension and duration of hypertension. level (n = 1), HbA1c (n = 3) [10,15,18,20,23,24,28,29,31], and two studies defined pre-DM using either one out of three parameters (fasting/postprandial plasma glucose level/HbA1c) [19,25]. The thresholds for defining pre-DM cases were also inconsistent, even within the same parameter. For example, pre-DM was defined as a HbA1c of ≥5.7% (≥39 mmol/mol) in five studies [13,18,19,22,23], with one study using a definition of ≥6.0% (≥42 mmol/mol) [20]. Of the nineteen studies that used fasting plasma glucose levels as the outcome definition, twelve used 6.1 mmol/L (110 mg/dL) as the threshold [7,12,14,15,16,17,19,25,26,27,28,30], while seven studies used 5.6 mmol/L (100 mg/dL) [11,13,21,24,29,31,32] for the diagnosis of pre-DM.

### 3.3. Predictors for Risk Prediction Tools

Predictors among the pre-DM risk prediction tools, and their frequencies of being included in a tool, are summarised in Figure 2. In general, non-laboratory-based pre-DM risk prediction tools included a median of six predictors (range: two to twelve). A total of 23 different predictors were identified among the 28 tools, including 15 clinical, 4 socio-demographic, and 4 lifestyle factors. The risk prediction tools tend to include more clinical factors than socio-demographic and lifestyle factors. Age was the most common predictor to predict pre-DM as it was included in all but one of the tools (96.4%) [14]. Other commonly included factors were obesity, measured by body mass index (BMI) or waist circumference (26 tools), family history of DM (19 tools), hypertension (18 tools), and sex (14 tools). The most commonly included lifestyle factor among the tools was exercise. Less common predictors included waist-to-height ratio [13], sleep duration [7], and macrosomia (applicable to a woman who had given birth to a child with an excessive birth weight) [30]. Notably, age, family history of DM, hypertension and obesity (represented by BMI and/or WC) were predictors included among all the tools that had been externally validated [10,11,12,14,15,17,22,24,25,26,27,29], indicating their robustness and reliability.

### 3.4. Methods for Tool Development

Logistic regression (LR) was used to develop the prediction tool in all but one of the studies [27]. Of the four studies (16.7%) that applied more than one development method (other than logistic regression) [7,16,22,31], all used machine learning (ML). No studies reported a significant difference in predictive performances in the tools developed by different methods. For instance, Dong et al. (2022) [7] developed two pre-DM and DM risk prediction models using LR and ML methods, and found similar performance results (AUROC: 0.81 and 0.82, respectively). Another study reported a slightly inferior performance of the classification tree model (AUROC: 0.69) when compared with the LR model (AUROC: 0.72) [16].

### 3.5. Performance of Risk Prediction Tools 

Performances of the risk prediction tools, when validated internally or externally, are summarised in Table 2. It was found that half of the studies validated their risk prediction tools using an external dataset [10,11,12,14,15,17,22,24,25,26,27,29], with three such studies having validated the tools with two or more external datasets [15,17,24]. Eight studies (33.3%) had validated their tools internally [7,16,18,20,21,23,30,31]. Among them, five studies used a proportionated dataset derived from the same source as the development dataset [7,16,18,30,31], two studies validated using bootstrapping methods [20,23] and one study randomly removed fifty participants from their development dataset in order to serve it as their validation sample [21]. The sample size of the validation dataset ranged from 50 to 66,108, with a median of 1987. Four studies (16.7%) did not perform any validation and only reported the tool performances that were generated during development [13,19,28,32].

The most frequently reported prediction performance measure was AUROC, but two studies did not report this for pre-DM prediction [14,27]. Existing tools showed mostly fair performances in internal validation, two studies that performed internal validation using the split-sample method yielded AUROCs above 0.8 [7,18]. On the other hand, the performance of the tools in external validation when available was more variable, with AUROCs ranging from 0.31 to 0.79, and mostly between 0.6 and 0.8. The prediction models developed by Wang et al. (2015) [15] performed poorly (AUROC: 0.31 and 0.50) when they were validated in an external dataset that was demographically different from the development dataset. It has been noted that the 95% confidence interval for AUROC was not reported in nine of the included studies (37.5%) [10,13,14,16,24,25,27,29,31], with two of the nine studies not presenting any information on AUROC [14,27]. 

Finally, sensitivities and specificities of the tools, acquired either during development or as a result of validation, together with their corresponding risk thresholds or cut-offs, were reported among all studies. Only around half of the studies reported assessment on the prediction tools’ goodness-of-fit, or the accuracy of the predicted risk against the observed risk, [7,11,12,15,17,18,19,21,22,23,24,25,30,32], by calibration plots or the Hosmer–Lemeshow test [33]. Furthermore, only one study [17] evaluated their prediction tools using more recent performance measures, such as the decision-curve analysis [34].

## 4. Discussion

This review identified 28 risk prediction tools that used only non-laboratory predictors to detect individuals with pre-DM from 24 published studies. The published prediction tools included similar predictors such as age, family history of DM, hypertension and obesity (represented by BMI and/or WC), despite the potential cultures and lifestyles differences of subjects from different study locations, supporting their robustness and reliability. The majority of existing non-laboratory-based tools (n = 26) had fair to good discrimination in case finding of pre-DM in the population that they were developed for. It was found that existing logistic regression (LR) and machine learning (ML) risk tools offered similar performance. However, pre-DM was inconsistently defined and the external validity of most tools was unclear.

It should be noted that these factors were also predictors of T2DM risks [35]. A family history of DM is a well-established risk factor for developing pre-DM [36], while associations between the age and DM risks have also been widely reported [37], but unfortunately these are not modifiable. Modifiable clinical factors (e.g., BMI, waist-to-hip ratio (WHR)) and lifestyle factors (e.g., number of hours of sleep and duration of physical activity) are particularly important because they offer potentials for intervention to prevent pre-DM and T2DM. It helps patient activation to emphasise the reversibility of pre-DM through healthy lifestyles. Interestingly, sleep hours was an important predictor in predictor tools published in recent years [7], but it was not considered in most studies, probably because the data were not available. Inadequate sleep duration has been associated with increased T2DM risks, which is likely due to the influence that sleep has on regulating endogenous hormones, such as testosterone and cortisol [6,38]. Indeed, by incorporating sleep hours as one of the predictors but without the inclusion of family history of DM, Dong et al. were able to obtain AUROCs over 0.8 for their tools in internal validation [7]. It is important to note that without a head-to-head comparison between existing tools, it is difficult to determine whether tools that include particular predictors offer statistically better predictions.

Although theoretically ML can develop more accurate prediction models by the inclusion of more complex parameter interactions, our review indicated that prediction tools developed by traditional LR and novel ML methods offered similar predictive performances in detecting pre-DM individuals. Sadek et al. further showed that ML models that were developed by different ML techniques offered no statistically significant difference in performance in an external dataset [22]. Consistent with the literature [39], the ML pre-DM prediction tools found in our review did not provide the weights that govern the interactions between predictors [22], while traditional LR models offered a comparatively higher interpretability on the interactions between predictors. Therefore, LR models could be more suitable for pre-DM risks prediction in real-world clinical practice, whilst ML approaches might be better suited for exploring and identifying novel predictors [40,41].

The majority of existing non-laboratory-based risk prediction tools for case finding of pre-DM showed satisfactory internal discrimination. As only half of the studies had validated their tools in an external dataset, the external performance of most existing tools could not be established. In general, a lower discriminatory ability was found in external validation. Notably, the Southern Chinese pre-DM risk prediction tool developed by Wang et al. showed good internal and external discrimination among datasets that shared similar socioeconomical characteristics as the development dataset, but not when their tools were applied to an external dataset that comprised Chinese people from Western China [15]. Such results suggest that the performance of a risk predication tool can be compromised when it is applied to a population living in a different environment to that of the development population. In order to offer an insightful evaluation on the tools’ performance, the validity and reliability in the population which they are intended to be used in as well as the appropriateness and representativeness of the dataset for external validation are important factors to be considered [42].

It was noted that some reporting inconsistency was found among the studies on non-laboratory-based pre-DM prediction tools. First, and most importantly, the case-definition of pre-DM was inconsistent among the studies, possibly due to a change in pre-DM definition by the American Diabetes Association in 2009 [43]. As a result, several studies used a fasting plasma glucose level of ≥6.1 mmol/L to define pre-DM/DM [44], while other studies used ≥5.6 mmol/L [45]. The case definition can have a significant effect on the discriminatory ability, sensitivity and specificity of the prediction tool. Second, the indictors of prediction performance varied widely among the studies. Although most of the studies reported the discriminatory ability by AUROC, many did not provide information on calibration. To focus only on discrimination could produce misleading predictions and potentially be detrimental during clinical decision-making processes [46]. In addition, a lack of 95% confidence intervals on the performance measures (e.g., AUROC, sensitivity, specificity, PPV, NPV) of existing tools was also noted. Overall, these reporting inconsistencies could hinder the generalisability and applicability of existing tools.

The strength of this review is that it included an up-to-date synthesis of the results from studies that used traditional and novel strategies for prediction tool development. Our findings also provide evidence to support the feasibility and efficacy of using only readily available non-laboratory predictors to facilitate case finding of pre-DM. Furthermore, the findings on the importance of sleep hours and duration of exercise can inform the development of interventions for the prevention and treatment of pre-DM. However, there are several limitations regarding our review that must be acknowledged. A meta-analysis was not performed on the included studies due to the large heterogeneity among the outcome measures and the reported performance indices. In addition, the inclusion of only studies published in the English language could have introduced bias and resulted in some pre-DM risk prediction tools being missed. 

## 5. Conclusions

This systematic review of 24 studies identified 28 non-laboratory-based pre-DM prediction tools. The most common predictors were age, family history of DM, hypertension and obesity measured by BMI and/or WC. Sleep hours and exercise duration were found to be important lifestyle predictors of pre-DM in more recent studies. Despite the difference in development methods, existing non-laboratory-based tools were mostly effective in the population that they were developed for. The generalisability of these tools was unclear as most of them had not been validated externally. External validation using datasets obtained from the intended target population should always be performed before application to practice for case-finding of pre-DM individuals. 

## Figures and Tables

**Figure 1 diagnostics-13-01294-f001:**
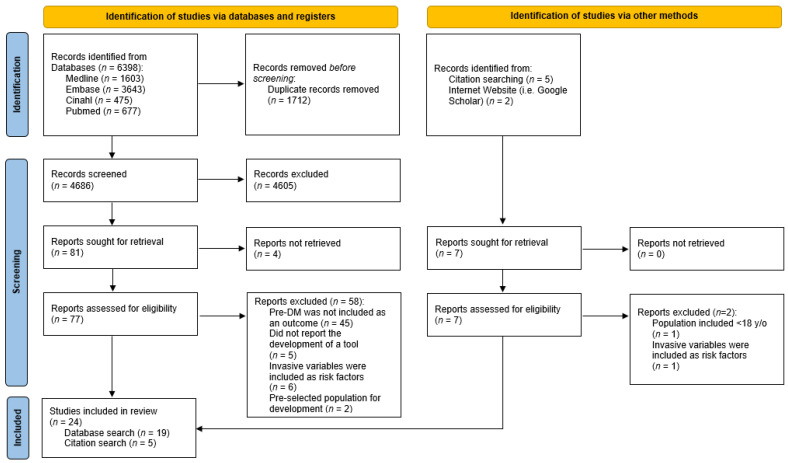
PRISMA flow diagram.

**Figure 2 diagnostics-13-01294-f002:**
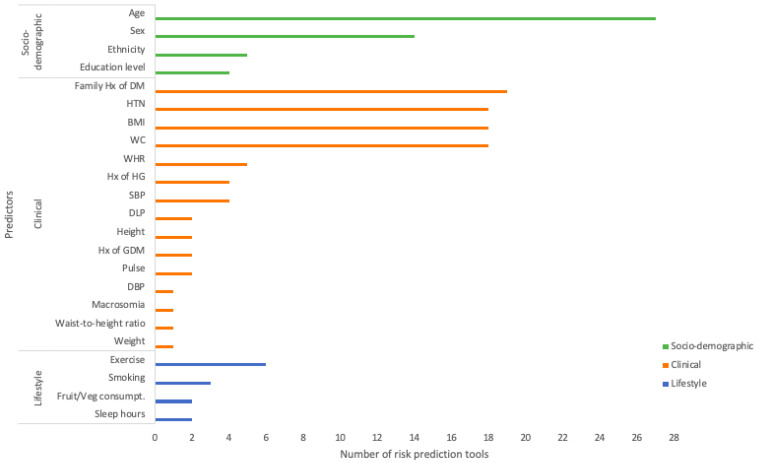
Non-laboratory-based predictors among existing prediction tools.

**Table 2 diagnostics-13-01294-t002:** Summary of validation performance of the pre-DM prediction tools reported by the included studies (*n* = 24).

Author, Year	Types of Validation (“I”/“E”)	Source of Validation Sample	Sample Size	Discriminative Performance (AUROC (95% CI))	Predictive Power (Sen. (95% CI), Spe. (95% CI), PPV (95% CI), NPV (95% CI))		
Abbas, 2021 *(LR Model)* [18]	I	Same dataset as development data (20/80 split)	1454	0.80 (0.78, 0.83)	0.86 (0.83, 0.89), 0.58 (0.55, 0.61), 0.50 (0.46, 0.53), 0.90 (0.87, 0.92)		
Bahijri, 2020 [19]	NA	No validation performed; performance data is from model development	-	0.76 (0.73, 0.79)	0.69, 0.69, 0.40, 0.88		
Barengo, 2017 *(IGR model)* [32]	NA	No validation performed; performance data is from model development	-	0.72 (0.69, 0.74)	0.57, 0.73, 0.58, 0.76		
Dong, 2022 [7]	I	Same dataset as development data (33/66 split)	619	LR: 0.81 (0.77, 0.85) ML: 0.82 (0.78, 0.86)	LR: 0.89, 0.62 0.31, 0.97	ML: 0.79, 0.74 0.36, 0.95	
Fu, 2014 *(Non-invasive model)* [10]	E	External community-based health survey dataset	1455	0.65	None reported for the non-invasive model		
Fujiati, 2017 [11]	E	External population-based health survey dataset	6933	0.65 (0.62, 0.67)	0.55 (0.51, 0.59), 0.66 (0.65, 0.67), 0.12 (0.11, 0.13), 0.94 (0.94, 0.95)		
Gao, 2010 *(PDM as the model outcome)* [12]	E	External population-based health survey dataset	4336	Men: 0.61 (0.58, 0.65) Women: 0.63 (0.61, 0.66)	Men: 0.86 (0.84, 0.87), 0.21 (0.19, 0.23), No PPV and NPV	Women: 0.76 (0.74, 0.77), 0.44 (0.42, 0.46), No PPV and NPV	
Gray, 2010 [26]	E	External population-based screening study dataset	3171	0.72 (0.69, 0.74)	0.81 (0.78, 0.84), 0.45 (0.43, 0.47), 0.29 (0.27, 0.31), 0.90 (0.88, 0.91)		
Gray, 2012 *(Validated by 2 definitions of outcome)* [25]	E	External population-based screening study dataset	3004	OGTT ^†^ as outcome: 0.69 HbA1c ^‡^ as outcome: 0.67	OGTT as outcome: 0.75 (0.71, 0.78), 0.52 (0.50, 0.54), 0.29 (0.26, 0.31), 0.89 (0.87, 0.91)	HbA1c as outcome: 0.75 (0.72, 0.78), 0.50 (0.48, 0.52), 0.37 (0.35, 0.40), 0.83 (0.81, 0.85)	
Gray, 2013 [24]	E	(1) External sampling by city-wide random digit dialling (2) External prospective 1-year follow-up data on the city-wide cohort	2131 1304	(1) 0.69 (2) 0.72	(1): 0.73 (0.69, 0.78), 0.56 (0.53, 0.58), 0.27 (0.24, 0.30), 0.90 (0.88, 0.92)	(2): 0.69 (0.63, 0.74), 0.63 (0.60, 0.67), 0.38 (0.34, 0.42), 0.86 (0.83, 0.89)	
Handlos, 2013 [20]	I	Same dataset as development data (split into 3 datasets based on original country)	(1) 2155; (2) 2446; (3) 1987	(1) 0.70 (0.67,0.72) (2) 0.70 (0.67,0.72) (3) 0.70 (0.67,0.73)	(1): 0.76 (0.72, 0.80), 0.50 (0.48, 0.52)	(2): 0.74 (0.70, 0.79), 0.54 (0.52, 0.57)	(3): 0.76 (0.72, 0.80), 0.52 (0.49, 0.54)
					No PPV and NPV	No PPV and NPV	No PPV and NPV
Henjum, 2022 [23]	I	Same dataset as development data	308	0.81		0.89, 0.65, 0.28, 0.97	
Hische, 2010 [27]	E	External opportunity sampling in healthcare centres in another city	1998	None reported	0.90, 0.32, 0.44, 0.85		
Koopman, 2008 [29]	E	External population-based health survey data	None reported	0.74	None reported for the external validation		
Memish, 2015 *(Dysglycemia model)* [21]	I	Same dataset as development data	50	0.68 (0.54, 0.82)	0.76 (0.55, 0.90), 0.68 (0.47, 0.84) No PPV and NPV		
Rajput, 2019 [13]	NA	No validation performed; performance data is from model development	-	0.79	0.84 (0.78, 0.90), 0.58 (0.55, 0.62) 0.31 (0.27, 0.34), 0.94 (0.92, 0.96)		
Robinson, 2011 [30]	I	Same dataset as development data (30/70 split)	1857	0.75 (0.73, 0.78)	0.70, 0.67, 0.35, 0.90		
Sadek, 2022 *(IGM model)* [22]	E	External population-based BioBank dataset	930	LR: 0.77 (0.74, 0.81) ML (1): 0.79 ML (2): 0.78 ML (3): 0.77 ML (4): 0.78	LR: 0.78, 0.69 0.45, 0.91 (from Appendix A)	ML (1–4): None reported	
Stiglic, 2018 *(IFG model)* [28]	NA	No validation performed; performance data is from model development	-	0.84 (0.81, 0.87)	0.73 (0.68, 0.79), 0.81 (0.74, 0.86), 0.60 (0.53, 0.67), 0.89 (0.87, 0.91) (from Appendix A)		
Tan, 2016 *(PDM model)* [14]	E	External opportunity sampling of individuals in the same region	83	None reported	0.92, 0.66 No PPV and NPV		
Wang, 2015 [15]	E	3 External population-based health survey datasets from different regions of China	(1) 1186; (2) 3162; (3) 1289	(1) Men: 0.75 (0.67, 0.83) Women: 0.77 (0.71, 0.83) (2) Men: 0.74 (0.61, 0.86) Women: 0.72 (0.65, 0.78) (3) Men: 0.31 (0.20, 0.43) Women: 0.50 (0.38, 0.61)	(1) Men: 0.73, 0.64, 0.13, 0.97 Women: 0.81, 0.60, 0.19, 0.96	(2) Men: 0.79, 0.51, 0.06, 0.99 Women: 0.89, 0.42, 0.05, 0.99	(3) Men: 0.31, 0.49, 0.02, 0.96 Women: 0.42, 0.59, 0.04, 0.96
Xin, 2010 *(PDM and T2DM model)* [16]	I	Same dataset as development data (50/50 split)	1130	LR: 0.72 Tree: 0.69	LR: None reported	Tree: 0.65, 0.73, 0.33, 0.91	
Yu, 2010 *(Classification scheme II)* [31]	I	Same dataset as development data (20/80 split)	983	LR: 0.73 SVM: 0.73	LR: None reported	SVM: 0.74, 0.63, 0.51, 0.82	
Yu, 2022 [17]	E	2 External population-based health survey datasets	(1) 1525; (2) 66,108	(1) 0.71 (0.63, 0.79) (2) 0.73 (0.73, 0.74)	None reported for the external validation		

Note: AUROC = Area Under the Receiver Operating Characteristic curve, “E” = External, IFG = Impaired Fasting Glucose, IGM = Impaired Glucose Metabolism, IGR = Impaired Glucose Regulation, “I” = Internal, LR = Logistic Regression, ML = Machine Learning, NA = Not applicable, NPV = Negative predictive value, PDM = Prediabetes, PPV = Positive predictive value, Sen. = Sensitivity, Spe. = Specificity, SVM = Support vector machines, T2DM = Type 2 Diabetes Mellitus. ^†^ = DM/PDM defined by FPG: ≥6.1 mmol/L or 2-h PG: ≥7.8 mmol/L, ^‡^ = DM/PDM defined by HbA1c: ≥6.0%

## Data Availability

No additional dataset was generated. All data extracted and synthesised was as summarised and reported in Table 1 and Table 2.

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
