# Peer review of "Non-Laboratory-Based Risk Prediction Tools for Undiagnosed Pre-Diabetes: A Systematic Review"

_diagnostics, 2023, doi:10.3390/diagnostics13071294_

Round 1

Reviewer 1 Report (Previous Reviewer 2)

I found the previously mentioned items to be corrected improved.

Reviewer 2 Report (Previous Reviewer 1)

I thank the authors for adequately responding to my comments. 

This manuscript is a resubmission of an earlier submission. The following is a list of the peer review reports and author responses from that submission.

Round 1

Reviewer 1 Report

This systematic review addresses an interesting topic, is well-written and well- organized and includes a clear description of its objectives and the methodology used.

However, I have two major and one minor points that I think should be addressed before publication.

Major:

1.     Almost a year has passed since the last search for relevant studies. To update the systematic review, the authors should perform an additional search and include any studies published after March 2022.

2.     The results lack a clear description of the factors which, according to the synthesis of available evidence, were shown to predict pre-DM with reliability and robustness. 

Minor:

1.     Lines 184-187: Please report fasting glucose values in mg/dl and HbA1c values in mmol/mol, as well.

Reviewer 2 Report

The manuscript titled:“Non-laboratory-based risk prediction tools for undiagnosed diabetes: A systematic review ” is a systematic review which aim is to identify important non-laboratory predictors of prediabetic status.

The review is reported in compliance with the PRISMA Checklist and PRISMA flowchart. To access the quality and risk-of-bias of the selected studies the authors used the CASP appraisal checklist. The details of the protocol were registered on the PROSPERO.

Based on the methodology the authors checked a total of 5388 studies of which finally 22 fulfilled the inclusion criteria. The authors detailly present the inclusion and exclusion criteria. Noteworthy they used two independent reviewers to screen the studies taken into analysis based on those criteria. The authors present the data and the results in a proper way. They show the character od analyzed studies, their quality and outcome of prediction tools. They are pointing out the limitations of the analysis resulting of heterogeneity of the studies. An example is using different definition of prediabetic status as expressed by HbA1c or fasting plasma glucose.

In the discussion section the authors discuss the results turning attention to many aspects of the results, both clinical and methodological (among them the lack of the external validity of many studies).

Reassuming the manuscript is worth publishing in Diagnostics